# Protein Phosphatases—A Touchy Enemy in the Battle Against Glioblastomas: A Review

**DOI:** 10.3390/cancers11020241

**Published:** 2019-02-19

**Authors:** Arata Tomiyama, Tatsuya Kobayashi, Kentaro Mori, Koichi Ichimura

**Affiliations:** 1Division of Brain Tumor Translational Research, National Cancer Center Research Institute, 5-1-1 Tsukiji, Chuo-ku, Tokyo 104-0045, Japan; opera58840428@gmail.com (T.K.); kichimur@ncc.go.jp (K.I.); 2Department of Neurosurgery, National Defense Medical College, 3-2 Namiki, Tokorozawa, Saitama 359-8513, Japan; kmori@ndmc.ac.jp; 3Department of Neurosurgery, Tokyo Women’s Medical University, 8-1 Kawadacho, Shinjuku-ku, Tokyo 162-8666, Japan

**Keywords:** protein phosphatase, glioblastoma, signaling, therapy

## Abstract

Glioblastoma (GBM) is the most common malignant tumor arising from brain parenchyma. Although many efforts have been made to develop therapies for GBM, the prognosis still remains poor, mainly because of the difficulty in total resection of the tumor mass from brain tissue and the resistance of the residual tumor against standard chemoradiotherapy. Therefore, novel adjuvant therapies are urgently needed. Recent genome-wide analyses of GBM cases have clarified molecular signaling mechanisms underlying GBM biology. However, results of clinical trials targeting phosphorylation-mediated signaling have been unsatisfactory to date. Protein phosphatases are enzymes that antagonize phosphorylation signaling by dephosphorylating phosphorylated signaling molecules. Recently, the critical roles of phosphatases in the regulation of oncogenic signaling in malignant tumor cells have been reported, and tumorigenic roles of deregulated phosphatases have been demonstrated in GBM. However, a detailed mechanism underlying phosphatase-mediated signaling transduction in the regulation of GBM has not been elucidated, and such information is necessary to apply phosphatases as a therapeutic target for GBM. This review highlights and summarizes the phosphatases that have crucial roles in the regulation of oncogenic signaling in GBM cells.

## 1. Introduction

Malignant gliomas are the most common primary intracranial malignant neoplasm arising from brain parenchyma. And, Glioblastoma (GBM) is the most malignant and aggressive form of gliomas (WHO grade Ⅳ) with the highly infiltrative phenotype, refractoriness, and genetical complexity [1,2]. Despite the efforts made over the past decades, therapeutic outcomes of GBM still remain poor [3,4], mainly because complete tumor mass resection is difficult because of its strong invasiveness into adjacent normal brain tissue and resistance by the residual tumor to standard chemoradiotherapy. Therefore, novel adjuvant therapies for GBM are urgently needed. The developments in high-throughput methods have allowed genome-wide analyses of malignant tumors, and landscapes of genetic mutations in GBMs have been reported [5,6,7,8,9,10,11,12,13,14]. In addition, molecular signaling networks involved in the regulation of GBM biology have been elucidated [5,6,15]. Concomitantly, clinical trials targeting signaling molecules identified in these studies have been started. However, most clinical trials have yielded unsatisfactory results [16,17].

Regulation of molecular signaling, including signaling related to tumor biology, generally involves post-translational modification of the signaling molecules. One of the most common modifications is phosphorylation. As a result of upstream signaling activation, kinases are activated and attach phosphate groups to certain amino-acid residues of their specific substrate molecules, resulting in downstream signaling activation. In GBM, kinases are often hyperactivated as a result of genetic alteration or enhanced upstream signaling activation, and, therefore, kinase hyperactivation has been focused on as a therapeutic target [5,6,18,19,20].

Protein phosphatases (PPPs), as negative regulators of phosphorylation-dependent signaling, remove phosphate groups of certain phosphorylated amino-acid residues of specific substrate molecules [21,22,23,24,25,26,27,28]. In tumor cells, most PPPs have been suggested to act as negative regulators of oncogenic signaling and, thus, to function as tumor suppressors, as oncogene-induced signal transduction is largely mediated by phosphorylation [22,23,29,30]. However, recent studies have demonstrated suppressed expression or activity of tumor-suppressive PPPs in certain tumors, including GBMs, and as such, PPPs might negatively regulate oncogene inhibition. However, our knowledge of the roles of PPPs in tumor signaling and oncogenic activity is still limited. In this review, we discuss the current knowledge on the roles of PPPs intertwined with kinase signaling in the regulation of GBM biology.

## 2. Protein Phosphatases

PPPs are classified as classical or atypical PPPs (Figure 1) [21,23,27]. Based on the amino-acid residue they dephosphorylate, classical and atypical PPPs are further grouped into protein serine/threonine phosphatases (PSPs), protein tyrosine phosphatases (PTPs), and dual-specificity phosphatases (DUSPs) (Figure 1) [21,27]. In classical PPPs, these groups are even more divided into subfamilies based on their chemical structures (Figure 1). Most of PPPs can suppress oncogenic signaling by dephosphorylating phosphorylated (activated) signaling molecules, such as mitogen-activated protein kinases (MAPKs), and are known to act as tumor suppressors in various tumors [21,31,32]. On the other hand, certain kinds of PPPs are known to be overexpressed in tumor cells and rather act as oncogenes [31,33,34,35,36,37]. Because both oncogenic signaling and tumor-suppressive signaling are simultaneously regulated by the same phosphatase, it is suggested that the oncogenic or tumor-suppressive function of PPPs in tumors probably depends on the balance between the two signaling effects of the PPPs. 

To develop clinical GBM therapies targeting PPPs, it is important to understand the detailed molecular mechanisms underlying both tumor-suppressive and oncogenic PPPs in GBM cells. In what follows, we discuss PPPs that are suggested to have major roles in GBM biology in detail.

### 2.1. Classical Protein Phosphatases

#### 2.1.1. *Protein*
*Serine*/*Threonine* Phosphatases

##### Protein Phosphatase 2A

As mentioned above, protein phosphatase 2A (PP2A) is one of the most major PSPs. PP2A is a heterotrimeric protein phosphatase complex which consists of the alpha (PPP2R1A) or beta (PPP2R1B) isoform of the structural A subunit, the alpha (PPP2CA) or beta (PPP2CB) isoform of the catalytic C subunit, and the regulatory B subunit. The A subunit and C subunit forms core heterodimer, and association of one of the multiple B subunits with the core dimer directs various substrate specificity (more than 60 combinations) of PP2A [39]. PP2A regulates various cellular signaling pathways, such as receptor tyrosine kinase (RTK) signaling, by dephosphorylating multiple substrates under physiological conditions, and ablation of PP2A expression or activity causes cardiovascular disorder, diabetes, and neurodegenerative disorder [26]. In cancer systems, involvement of genetic, epigenetic, or post-translational modification-mediated dysregulation of PP2A expression or activity in tumorigenesis are suggested, and dysregulated PP2A tumor cells cause an increase in cellular proliferation, formation of resistance against drug or irradiation, or impairment of tumor immunity [26,40,41,42,43]. However, the genetic alteration of PP2A subunits-encoding genes in GBMs are rare (about less than 1%) in The Cancer Genome Atlas (TCGA) datasets [5,43]. One of the mechanisms which is suggested to induce non-genetic dysregulation of PP2A in GBM is hyperactivation of RTKs, such as epidermal growth factor receptor (EGFR), by genetic alteration frequently observed in GBMs [5,6]. In a certain series of malignant tumors with RTK hyperactivation, downregulation of PP2A expression or activity has been reported, which would possibly relieve PP2A-mediated suppression of downstream signaling of RTK, resulting in further activation of RTK-mediated signaling [26,44,45,46]. In line herewith, downregulated expression of PP2A subunits—without genetic alteration—has been observed in glioma tissue [47,48]. And direct or indirect inhibition of PP2A resulted in enhanced oncogenic property of glioma cells [43,49,50,51], suggesting a role of PP2A as a tumor suppressor in GBMs. As the other non-genetic regulatory mechanisms of PP2A activity, the molecules which negatively regulate PP2A activity are also crucial. Among this group of proteins, cancerous inhibitor of PP2A (CIP2A), protein phosphatase methylesterase-1 (PME-1), and SE translocation (SET) oncoprotein, are well-characterized and known to downregulate PP2A activity by different biological processes [26]. CIP2A directly associates with and blocks the B56 regulatory subunits of PP2A complex [52], and importantly, high expression of CIP2A is correlated with overexpression of EGFR in the certain cancer systems [44,45,46]. PME-1 suppresses PP2Ac activity by the removal of metal ions from PP2Ac catalytic core and demethylation of the C-terminal lesion of PP2Ac, whereas SET directly associates and blocks the catalytic core of PP2Ac [53,54]. In GBMs, in vitro experiments revealed the possible role of PME-1 in the formation of GBM cell resistance against Ca^2+^/calmodulin-dependent protein kinase inhibitor (H7), PI3K inhibitor (LY29644), and multi-RTKs inhibitor (sunitinib). These knowledges suggest not only expressional but also enzymatic inhibition of PP2A in GBM cells would be important for the maintenance of GBM malignancy, and the possible role of PP2A reactivation as the therapeutic strategy of GBM would also be considered (see below chapter 3.1. On the contrary, PP2A has also been suggested as a potent therapeutic target for GBMs. Treatment with PP2A inhibitor okadaic acid alone, without concomitant use of genotoxins, triggered mitotic cell death of GBM cells [55]. Treatment of GBM stem cells with a PP2A inhibitor LB100 resulted in induction of differentiation or cell death via dysregulation of nuclear receptor corepressor [56]. Treatment of GBM cells with the c-Jun N-terminal kinase (JNK) activator anisomycin induced cell death via suppression of PP2A subunit expression [57]. Pharmacological inhibition of PP2A activity by LB-1.2, LB-100, or Microcystin-LR under irradiation or genotoxin treatment resulted in enhanced cell death induction in GBM cells [58,59,60]. As mentioned above, these pro-survival effects of PP2A in GBM cell death might be due to the alteration of the balance between cell death-inducing and cell survival-inducing signaling under PP2A inhibition. In fact, PP2A regulates both cell death-inducing and cell survival-inducing signaling simultaneously in multiple sites of programmed cell death cascades (Figure 2) [26,38,61,62,63]. Collectively, these evidences suggest the potential of PP2A as a therapeutic target for GBMs not only through suppression but also through upregulation of its activity or expression.

##### Protein Phosphatase 4

Protein phosphatase 4 (PP4) is a member of the type 2A PSP family, which includes PP2A, PP4, and protein phosphatase 6 (PP6) [64,65]. These PSPs have approximately 60% sequence similarity, but they have different biological functions. In normal conditions, PP4 is reported to regulate the DNA-damage response as well as NF-kappaB and mTOR functions [66]. In several cancers, enhanced expression of PP4 subunits is observed in the cancer tissue and is suggested as a marker of poor prognosis in these cases [67,68,69,70]. Although alteration of *PPP4C* gene, which encodes the catalytic subunit of PP4 (PP4C), in the TCGA GBM dataset was not confirmed [5], high expression of PP4C was observed in a series of GBM cases, and the PP4C expression level was negatively correlated with prognosis [67]. In addition, knockdown of PP4C in cultured GBM cells resulted in a reduction in their oncogenic property [67]. Although the positive evidence of PP4 in GBM biology as noted above has been reported, the exact role of PP4 in GBM biology has not yet been elucidated, and more evidence of its potential as a novel therapeutic is needed.

##### Protein Phosphatase 5

Protein phosphatase 5 (PP5) is one of the classical PSPs encoded by *PPP5C* and highly expressed in central nervous system and neurons [71,72]. Physiologically, PP5 regulates cellular survival, differentiation, DNA damage repair, migration, and proliferation [73]. On the other hand, high expression of PP5 in the breast cancer or osteosarcoma tissue is observed [74,75]. However, no direct evidence about linking between aberrant expression of PP5 with tumorigenesis is reported yet. In GBMs, in vitro experiments revealed positive contribution of PP5 in regulation of GBM cell growth and migration [76]. Therefore, to investigate further about expression level of PP5 in GBM tissue and the detailed regulatory roles of PP5 in GBM oncogenicity would be suggested as meaningful.

##### Protein Phosphatase 6

Like PP4, Protein phosphatase 6 (PP6) is a member of the type 2A PSP family. It regulates various physiological processes, such as anti-inflammation, cell-cycle regulation, DNA-damage repair, and lymphocyte development. Furthermore, it is reportedly involved in the regulation of tumorigenesis. In melanomas, PP6 has been shown to act as a tumor suppressor [65]. On the other hand, anti-tumor activity of PP6 is reported in cervical cell carcinoma cell [77,78]. In GBMs, the PP6 catalytic subunit (PP6c) is overexpressed in around 0.5% of the TCGA GBM dataset [5], and siRNA knockdown of PP6c suppressed DNA-dependent protein kinase activity, resulting in an enhanced response of GBM cells to irradiation treatment in vitro and in vivo [64]. The representative PP2A inhibitor, okadaic acid, also inhibits PP6 activity [65], and okadaic acid has a tumor-suppressive effect in GBM cells [55,56], which is likely through suppression of not only PP2A but also PP6 activity. These evidences suggest PP6 rather contributes as an oncogene in GBMs, but the accumulation of further evidences is necessary for clarifying the exact roles of PP6 in GBM biology as well as PP4.

#### 2.1.2. Protein Tyrosine Phosphatases

##### Tyrosine-Protein Phosphatase Non-Receptor Type 6

Tyrosine-protein phosphatase non-receptor type 6 (PTPN6; also known as SHP1) is a PTP primarily expressed in hematopoietic cells in normal tissue and regulates hematopoietic signaling, such as tyrosine-protein kinase Lyn-mediated pathway. In a series of glioma cases, high expression of SHP1 in glioma tissue was associated with poor prognosis [79]. Furthermore, in vitro study revealed expression of SHP1 in GBM cells resulted in an increase of chemoresistance, and it was also demonstrated that the expression level of SHP1 in glioma tissue was regulated by SHP1 promoter methylation status [79]. These results suggest SHP1 would have rather oncogenic roles in GBM cells, and epigenetic machinery is a key mechanism for regulation of SHP1 expression in GBMs.

##### Tyrosine-Protein Phosphatase Non-Receptor Type 11

Tyrosine-protein phosphatase non-receptor type 11 (PTPN11; also known as PTP-1D, PTP-2C, or SHP2) is a PTP activated mainly by RTKs and is one of the representative PTPs that have oncogenic roles in various cancers [24,33,80,81,82,83,84,85]. Dysregulation of SHP2 function because of germline mutations is involved in the pathogenesis of hereditary diseases, such as Noonan syndrome and Leopard syndrome, as well as in oncogenesis and malignancy of neoplasms [24,33,80,81,82,83,84,85,86,87]. The major signaling pathway regulated by SHP2 is the Ras-Raf-ERK (MAPK) cascade [33,80,81,82,83,84]. Both the hereditary diseases and oncogenic signaling triggered by dysregulation of SHP2 are known to be mediated by alterations of MAPK cascade activity [33,81,82,85,86,87]. Although SHP2 is reported to negatively regulate proto-oncogenes, such as STAT3, by dephosphorylation [88,89], SHP2 as a positive regulator of the MAPK cascade is reported to contribute as a proto-oncogene. There is evidence that SHP2 activates the MAPK cascade via adapter molecules, such as growth factor receptor-bound protein 2—although the detailed regulatory mechanism remains unknown—and through dephosphorylation of Ras [33,80,83]. In glioma cells, although MAPK cascade activation by oncogenic activation of other molecules is frequently reported, genetic alteration or enhanced activation of SHP2 is not common [5,6,13]. This is probably because SHP2 induces downregulation of other signaling factors, such as signal transducer and activator of transcription 3 (STAT3), which is also essential for glioma maintenance, and overactivation of SHP2 rather leads to repression of tumor growth, even when the MAPK pathway is activated. On the other hand, SHP2 has been reported to play an essential role in oncogenic signal transduction of EGFRviii, an activation mutation of EGFR frequently observed in GBMs [90]. In addition, reduced expression of SHP2 resulted in augmented radiosensitivity in glioma cells [91], suggesting that SHP2 has potential as a therapeutic target for GBM, even without oncogenic activation or overexpression. In line with this, the efficacy of SHP2 inhibitors in GBM treatment has been exhibited (see below chapter 3.2.). 

##### Tyrosine-Protein Phosphatase Non-Receptor Type 13

Tyrosine-protein phosphatase non-receptor type 13 (PTPN13 or PTPL1) is a PTP that regulates various cellular functions, such as proliferation and differentiation, via the Ras-ERK cascade or the Rho-associated protein kinase (ROCK) pathway [24,92,93,94,95]. PTPN13 has shown both tumor-suppressive and oncogenic effects in several cancers [22,93,95,96,97,98,99]. As for GBMs, PTPTN13 is overexpressed in GBM cells compared with normal cells in GBM tissue [22,37]. An in vitro study revealed PTPN13 directly interacts with and dephosphorylates tyrosine-phosphorylated FAS, a cell membrane-localizing death receptor activated by interaction with FASL ligand, and blocks FASL-dependent cell death in GBM cells [37]. Because FAS and FASL-mediated cell death machinery are involved in irradiation- or genotoxin-induced cell death of GBM cells [37,100,101], these findings suggest that PTPN13 acts as an oncogene in GBM cells by inhibiting therapy-induced GBM cell death.

##### Receptor-Type Tyrosine-Protein Phosphatase Delta 

Receptor-type tyrosine-protein phosphatase delta (PTPRD) is one of the receptor-type tyrosine kinases (RTPTPs). PTPRD is encoded by the *PTPRD* gene and is inactivated in about 2 to 3% of GBM cases in the TCGA dataset [5]. Physiologically, PTPRD regulates hippocampal memory by promoting neurite outgrowth or axon guidance in the central nervous system [102], and mutation of *PTPRD* triggers craniosynostosis, hearing loss, or intellectual disability [103,104]. In GBMs, suppression of PTPRD expression by missense or nonsense mutation or promoter hypermethylation of *PTPRD* is frequently observed [105,106]. Forced expression of PTPRD in cultured GBM cells resulted in growth arrest or cell death that might be because of inhibition of activation of STAT3, one of the substrates of PTPRD [105,107]. These findings suggest that PTPRD is a tumor suppressor in GBM, but when genetically or epigenetically inactivated, might contribute to tumorigenesis in GBM. 

##### Receptor-Type Tyrosine-Protein Phosphatase Mu

Receptor-type tyrosine-protein phosphatase mu (PTPRM or RTPTPμ) is RPTPT expressed in endothelial, glial, and neuronal cells. PTPRM regulates various biological process, such as cell growth, differentiation, and mitosis. And importantly, PTPRM positively regulates cell–cell adhesion by association with another PTPRM expressed at adjacent cells via homophilic binding [108]. The role as the negative regulator of cancer biology is indicated [109,110]. In GBMs, decreased expression of PTPRM is reported in GBM tissue [111], and suppression of PTPRM expression in GBM cells resulted in enhancement of migration in vitro [111], suggesting decreased PTPRM expression contributes to tumorigenicity of GBM by augmentation of GBM cell migration. However, inhibition of PTPRM phosphatase activity by antagonizing peptide demonstrated inhibition of GBM cell migration [112]. Therefore, the therapeutic roles of PTPRM in GBM treatment should be further confirmed carefully.

##### Receptor-Type Tyrosine-Protein Phosphatase Zeta

Receptor-type tyrosine-protein phosphatase zeta (PTPRZ) is an RTPTP encoded by *PTPRZ1* [113,114]. In the central nervous system, PTPRZ regulates neurotransmission, endocytic transportation, and synapse motility, thereby controlling hippocampal memory under the physiological condition. Through alternative splicing, three variants of PTPRZ are generated: a long form, a short form, and phosphacan, which is the extracellular domain of PTPRZ and is secreted to the extracellular space [115,116]. Phosphacan is expressed mainly in matured glia in the adult human brain, whereas the other two variants are expressed mainly in glial precursors [115]. In GBM tissue, all variants are expressed, and high-level expression of PTPRZ is reported [117,118]. Importantly, single-cell RNA sequencing analysis of primary GBM cases revealed that PTPRZ positively regulates stemness of GBM cells [119]. In addition, PTPRZ knockdown resulted in a suppression of migration and tumor growth of GBM cells [117,120,121]. Although the exact role of the intrinsic phosphatase activity of PTPRZ in the regulation of GBM biology remains unclear, recent evidence revealed that re-expression of the extracellular domain in GBM cell lines after knockdown of PTPRZ resultthe ed in rescue of the migration, but not proliferation, in these cells [120]. Enhanced secretion of pleiotrophin, one of the extracellular ligands of PTPRZ abundantly expressed in GBM tissue, from tumor-associated macrophages in GBM tissue promotes tumor growth of GBM stem cells [122]. The biological and therapeutic roles of the extracellular domain, as well as the intracellular PTP domain of PTPRZ. should be further investigated to elucidate the oncogenic role of PTPRZ in GBM. Recently, the inhibitors against PTPRZ, SCB4380 and NAZ2329, have been developed, and efficacy of these inhibitors in tumorigenicity of GBM cells was demonstrated (see below chapter 3.3).

#### 2.1.3. Dual-Specificity Phosphatases

##### Phosphatase and Tensin Homolog 

Phosphatase and tensin homolog (PTEN) is one of the best known DUSPs in gliomas, because inactivation of PTEN by deletion or mutation was discovered in 41% of the TCGA GBM dataset [5]. In physiological conditions, PTEN catalyzes the dephosphorylation of phosphatidylinositol (3,4,5)-trisphosphate to phosphatidylinositol (4,5)-bisphosphate, which inhibits phosphatidylinositol-4,5-bisphosphate 3-kinase signaling [29,123,124]. This reaction consequently blocks RTK-mediated survival signaling, resulting in tumor suppression. In gliomas, PTEN has been highlighted for its tumorigenic role induced by genetic inactivation through chromosome 10 deletion. In addition to genetic inactivation of PTEN, recent evidence demonstrates that miR-26a, which is highly expressed in GBM tissues, targets PTEN and suppresses PTEN expression in GBMs with monoallelic PTEN deletion, resulting in further inhibition of PTEN [125]. *PTEN* promoter methylation-mediated PTEN silencing has also been detected in GBMs [126]. Because overactivation of RTKs, such as EGFR, by chromosomal mutations also frequently occurs in gliomas [5,6,15,127,128], it is suggested that PTEN inactivation results in enhanced RTK-mediated oncogenic signaling and. thus, to oncogenicity in glioma cells. This also implies that the phosphatidylinositol-4,5-bisphosphate 3-kinase pathway might be an effective therapeutic target for gliomas, although direct targeting of PTEN would be difficult. Interestingly, a recent finding demonstrated PTEN is exocytically secreted by exosomes from GBM cells and suppresses Akt activity of its recipient cells in vitro [129], and exosomal microRNA (miR)-21 and miR-26a enriched in GBM patients as the circulating microRNA suppress PTEN expression [130], suggesting the profound roles of exosomes in regulation of GBMs biology via modulation of PTEN expression. Therefore, understanding these epigenetical or exosomal regulations of PTEN are also suggested to be important for targeting PTEN-mediated signaling pathway in GBMs treatment.

##### Dual-Specificity Phosphatase 1/Mitogen-Activated Protein Kinase Phosphatase 1 

Most DUSP family members regulate the phosphorylation (activation) status and subcellular localization of MAPKs [22,25,28,131]. Dephosphorylation of MAPKs by DUSPs usually results in the suppression of MAPK signaling induced by, e.g., RTKs [22,25,28,131]. Dual-specificity phosphatase 1 (DUSP1)/mitogen-activated protein kinase phosphatase 1 (MKP-1) is encoded by *DUSP1*, and DUSP1 expression is transcriptionally regulated by tumor suppressor p53 [132], which is frequently inactivated by genetic alteration in GBMs [5,6]. Thus, suppression of DUSP1 expression by p53 mutation would contribute to GBM tumorigenicity via upregulated MAPK signaling activation. Interestingly, one report demonstrated that DUSP1 expression is higher in GBM than in normal cells in glioma tissue [133]. Knockdown of DUSP1 in GBM cells resulted in enhanced induction of cell death by genotoxins, which was probably induced by JNK hyperactivation [134]. These findings suggest the potential of DUSP1 as a potent therapeutic target for GBMs, and that DUSP1 has both tumor-suppressive and oncogenic roles in GBMs, likely dependent on its expression level.

##### Dual-Specificity Phosphatase 4/Mitogen-Activated Protein Kinase Phosphatase 2 

Dual-specificity phosphatase 4 (DUSP4)/mitogen-activated protein kinase phosphatase 2 (MKP-2) is encoded by *DUSP4* and is widely expressed in the nucleus in various tissues. Like other DUSPs, DUSP4 dephosphorylates MAPKs, such as ERK1, ERK2, and JNK, and regulates proliferation and differentiation [22,131]. In cancers, DUSP4 expression is either up- or downregulated [135,136,137,138,139], and in GBM tissues, DUSP4 is generally downregulated [140]. This downregulation of DUSP4 expression is often associated with and, thus, likely caused by *DUSP4* promoter methylation [140,141]. Importantly, in GBMs, *DUSP4* promoter methylation is reported to be associated with alteration of the *IDH1* gene encoding isocitrate dehydrogenase 1 (IDH1) [140], which catalyzes the oxidative decarboxylation of isocitrate to 2-oxoglutarate [142]. *IDH1* mutation is one of the most important genetic alteration in gliomas, which is not only necessary for molecular diagnosis of glioma cases, but is also essential for understanding glioma biology [143]. *IDH1* mutations in gliomas are gain-of-function mutations that lead to the generation of the oncometabolite 2-hydroxyglutarate instead of 2-oxoglutarate from isocitrate, which induces a wide range of epigenetic dysregulation collectively termed “glioma CpG island methylator phenotype” [144]. The presence of *IDH1* mutation is one of the most important determining factors for molecular diagnosis in the WHO 2016 classification of gliomas [145,146]. Therefore, downregulation of DUSP4 expression in gliomas is suggested to be caused by epigenetic regulation triggered by *IDH1* mutation and to contribute to glioma tumorigenesis through enhancement of MAPK signaling. Accordingly, the DUSP4 expression level might serve as a predictor of *IDH1* mutation, which currently is crucial for the molecular diagnosis of gliomas.

##### Dual-Specificity Phosphatase 26/Mitogen-Activated Protein Kinase Phosphatase 8

Dual-specificity phosphatase 26 (DUSP26) is encoded by *DUSP26* and dephosphorylates MAPKs, including ERK, JNK, and p38; however, its physiological function remains unclear. In tumor cells, DUSP26 is known to act as both oncogene and tumor suppressor. A recent study revealed that the oncogenic activity of DUSP26 depends on dephosphorylation and inactivation of tumor suppressor p53 [147]. In GBMs, DUSP26 expression is downregulated, and lower expression of DUSP26 predicts poor prognosis in GBM patients [148]. Another study reported downregulation of DUSP26 expression GBM tissues and DUSP26-mediated enhancement of cell–cell adhesion, which suppressed the invasive phenotype of GBM cells [149]. Collectively, these findings indicate that DUSP26 might act as a tumor suppressor in GBMs, and suppression of activity or expression of DUSP26 might contribute to tumorigenesis of GBMs. 

##### Phosphatase of Regenerating Liver 3

Phosphatase of regenerating liver 3 (PRL-3) is a DUSP that positively regulates various cellular signaling factors, such as PI3K, Src, and Rho [35,150,151,152]. The C’-terminal region of PRL-3 is known to be prenylated, which might contribute to the intracellular membrane localization of PRL-3. The exact PRL-3-dependent regulatory molecular signaling and substrates of PRL-3 in normal cells are yet to be elucidated. It has been reported that PRL-3 contributes to growth induction, invasion, and metastasis of tumor cells [152,153,154,155]. In glioma cells, PRL-3 induces matrix metalloprotease expression via ERK, JNK, or other machinery, and this has been suggested to trigger invasion and metastasis [22,156,157,158]. More importantly, the PRL-3 expression level is elevated according to historical malignancy grade, and evidence demonstrates that PRL-3 expression is inversely correlated with the prognosis of GBM patients [22,156,157]. Based on these findings, it is suggested PRL-3 might be involved in the regulation of GBM malignancy; however, details on the molecular machinery behind PRL-3-mediated GBM oncogenesis required further investigation.

##### Cyclin-Dependent Kinase Inhibitor 3

Cyclin-dependent kinase inhibitor 3 (CDKN3, or KAP) is a DUSP that physiologically dephosphorylates cyclin-dependent kinase 2 (CDK2) and induces mitotic G1-S arrest [159,160]. In several cancers, KAP is reported to contribute as either a tumor suppressor or oncogene by de-phosphorylation of other targets. Although alteration of *CDKN3* gene, which encodes KAP, in the TCGA GBM dataset was not confirmed [5], KAP expression was increased, and a high KAP expression was associated with poor prognosis in a series of GBM cases [161]; however, KAP proteins expressed in GBMs were aberrantly spliced and acted dominant-negatively against wild-type KAP in the same cases [161]. An in vitro study revealed that wild-type KAP inhibits GBM cell migration in a CDC2-dependent manner [161]. In addition, a recent study revealed that KAP-ROCK2 pathway inhibition through overexpression of miR-26a which directly targets KAP and is often amplified in GBMs augments CDK2 activation in GBMs, resulting in disease progression [162]. Therefore, it is suggested downregulation of KAP activity by aberrant splicing or overexpression of microRNA facilitates GBM cell oncogenicity at least in part by promoting migration in a CDC2-dependent manner.

##### CDC25 Family

The cell division cycle 25 (CDC25) family of DUSPs is commonly involved in mitotic entry in the cell cycle through dephosphorylation and activation of cyclin-dependent kinases (CDKs). In many cancers, the upregulated expression of CDC25s and its role in tumor progression by accelerating proliferation has been confirmed. All three CDC25 variants (CDC25A, B, and C) are involved in the regulation of GBM biology. Although genetic alterations of the genes encoding CDC25s are not observed in the TCGA GBM dataset, in a certain series of human glioma samples, expression of the cell-cycle marker Ki67 is increased in parallel with increased CDC25A expression, and dephosphorylation of PKM2 by CDC25A induces glycolytic metabolism, resulting in the Warburg effect and tumorigenicity of GBM cells [163,164]. On the other hand, suppression of CDC25A expression during ionizing radiation in GBM cells resulted in enhanced invasion, and upregulation of CDC25A expression upon irradiation augmented ionizing radiation-induced GBM cell death [165,166]. These findings suggest that CDC25A is involved in positive regulation of GBM biology. However, CDC25A might also conversely contribute to radioresistance of GBMs. In the case of CDC25B, increased CDC25B expression in GBM cases is reported to associate with worsening of histological malignancy and poor prognosis [22,167,168]. Inhibition of the expression and activity of forkhead box protein M1 (FOXM1), an upstream regulator of CDC25B, by a natural compound plumbagin resulted in suppression of GBM cell growth [169,170]. Although one report indicated that protein expression of CDC25B in GBM tissue is not upregulated in cases with chemo- or ratio-resistance [171], these evidences suggest and an oncogenic rather than a tumor-suppressive role of CDC25B. In the case of CDC25C, genetic alteration of CDC25C and correlation of expression or activation of CDC25C with malignancy or prognosis is not common in GBMs; however, indirect pharmacological inhibition of CDC25C expression by demethocurcumin or ansamycins has been suggested to be effective in GBMs [172,173]. Collectively, the data suggest that all CDC variants act as proto-oncogenes. Although direct inhibition of CDCs is under development for clinical use, CDCs are strong candidates as a novel therapeutic target for GBMs. 

### 2.2. Atypical Protein Phosphatases

#### Eyes Absent Transcriptional Coactivator and Phosphatase Homolog 2

The eyes absent (EYA) family (EYA1–4) is a group of atypical PPPs that function as transcriptional cofactors as well as phosphatases. As for the physiological functions of the EYA family, they are known to be involved in organ development by regulating RTK, transforming growth factor (TGF) and Hedgehog-mediated signaling [23,174,175,176]. In addition, EYAs contribute to DNA repair [23,177,178,179] and regulate cancer proliferation and metastasis in various cancers [23,180,181]. EYA2 has a notable function in the regulation of cancer biology and serves as a prognostic marker in cancer [181,182]. In gliomas, including GBMs, EYA2 is overexpressed in glioma cells in glioma tissues compared with normal cells and is associated with histological grading. In the TCGA GBM dataset, amplification of EYA2-encoding gene *EYA2* is observed in around 0.7% of the total cases [5]. An in vitro study revealed that EYA2 positively regulates GBM cell invasion by enhancing matrix metalloprotease 9 expression and proliferation [183]. Thus, although the knowledge on EYA2 in the context of GBM is still limited, EYA2 might serve as a therapeutic target for GBM, as has been suggested for other tumors. 

## 3. Phosphatase Targeting in GBM

Recently, to establish the therapy targeting phosphatases based on the biological features of each phosphatase upon neoplasms, the small molecule inhibitors and activators against some phosphatases have been developed as the antitumor agents. The summary of these agents is shown in Table 1.

### 3.1. PP2A Inhibitors and Activators

As noted above, PP2A works as either oncogenic protein or tumor suppressor which is dependent on the type harboring tumor, and the inhibitor and activator have been the focused as novel anticancer drug against various malignancies previously. LB-100, a water-soluble small molecule inhibitor of PP2A, and its lipid-soluble derivative LB-102 have been shown antitumor effects against various tumors [184]. And importantly, LB-102 potentiated the effect of genotoxins without increase of side effects in the mouse xenograft model of human GBM cell lines [58]. Based on these evidences, phase 2 study of LB-100 against relapsed GBMs has already been started (https://clinicaltrials.gov/ct2/show/NCT03027388). However, okadaic acid-induced dormancy of GBM stem-like cells was also reported [208]. This means PP2A inhibitor might also have the potency to induce resistance against treatment-induced GBM cell death. On the other hand, the usefulness of PP2A activators has also been validated in tumor treatment. Phenothiazine, a tricyclic neuroleptic, induces PP2A-mediated apoptosis in leukemia cells via targeting A subunit of PP2A [209]. OP440, a peptidyl inhibitor of PP2A targeting SET, also demonstrated antitumor activity against leukemia cells [194]. Anticancer activity of FTY720, a sphingosine-based classical PP2A reactivator which targets SET, against various cancer systems has been demonstrated [185]. In GBMs, in vitro study demonstrated apoptosis-inducing activity of FTY720 against human glioma cell lines [186]. In addition, ceramides are also known to inhibit SET, and anti-inflammatory drug indomethacin is reported to induce upregulation of PP2A activity and PP2A-mediated GBM cell death by an increase of intracellular ceramides [210]. Currently, a series of orally bioavailable small molecule activators of PP2A (SMAP), which have been obtained by reengineering of tricyclic neuroleptics and targets structural A subunit of PP2A, are highlighted as the potent PP2A activators [187]. Efficacy of DT-061, a lead molecule of SMAP, in treatment of rat sarcoma proto-oncogene (RAS)- or MAPK-driven tumors, such as lung cancers or prostate cancers, has been confirmed [41,188]; however, the evidences about the effectiveness of SMAP in the treatment of GBMs is not reported yet. Taken together, the potencies of both PP2A inhibitors and PP2A activators as the antitumor agents against GBMs have been shown especially in association with recent progress in drug development, and it is also suggested a further continuation of investigation and improvement about these PP2A-targeted drugs is still necessary for clinical application.

### 3.2. SHP2 Inhibitors and Activators

As mentioned above, not only tumor-suppressive roles but tumorigenic functions of SHP2 have been suggested in various cancer systems. Therefore, SHP2 inhibitors, as well as SHP2 activators, have been developed as the anticancer drugs. The classical SHP2 inhibitors, such as NSC-87877 or PHPS1 which targets the catalytic domain of SHP2 [211,212], demonstrated antitumor effect against GBM cells in the cell-based studies. Cryptotanshinone, another SHP2 inhibitor extracted from natural plants [189], has also been reported to show tumor-suppressive effects upon GBM cells via suppression of STAT3 activation [190,191]. However, these active site-targeted SHP2 inhibitors or cryptotanshinone are also revealed to have the potency to act on multiple molecular targets [192,193]. Recently, RMC-4550, SHP099, and TNO155, the allosteric inhibitors of SHP2, have been developed, and these allosteric SHP2 inhibitors block SHP2 activity by lower concentration and higher specificity [195,196,197]. Notably, phase Ⅰ clinical trial of TNO155 as the therapeutic agent against solid tumors has already started (https://clinicaltrials.gov/ct2/show/NCT03114319); however, the efficacy of these allosteric inhibitors of SHP2 in GBM treatment has not yet been well-proven. On the other hand, the activator of SHP2, geranylnaringenin (CG902), is reported to inhibit proliferation of the prostate cell line via SHP2-dependent STAT3 suppression [213]. Although the efficacy of geranylnaringenin in GBM therapy is not validated yet, the possible role of geranylnaringenin in GBM treatment would be suggested because STAT3 is known to play a crucial role in the maintenance of GBM biology [214].

### 3.3. PTPRZ Inhibitor

As mentioned above, suppression of PTPRZ expression resulted in inhibition of GBM cell oncogenicity. SCB4380, a cell-impermeable small molecule PTPRZ inhibitor which also potentially blocks PTPRG activity, suppressed migration, proliferation, and tumorigenicity of the Rat GBM cell line by liposome-mediated intracellular loading [198]. NAZ2329, another cell-permeable allosteric inhibitor of both PTPRZ and PTPRG, demonstrated suppression of stem cell-like properties and tumorigenicity in GBM cells [199]. These results suggest the potency of PTPRZ inhibitors as the effective GBM-treating agents; however, especially from the viewpoint of target selectivity and drug delivery, further development of PTPRZ inhibitors would be necessary for actual clinical use.

### 3.4. DUSP1 Inhibitors

The various inhibitors targeting certain DUSPs (DUSP1, DUSP6, and DUSP26) have been developed for cancer therapy [25]. Among them, triptolide, a plant-derived DUSP1 inhibitor which suppresses DUSP1 expression, suppressed proliferation and invasion of GBM cells in vitro [200]. (E)-2-benzylidene-5-bromo-3-(cyclohexylamino)-2,3-dihydro-1H-inden-1-one (BCI), an allosteric inhibitor of DUSP1/6 was recently developed [201] and demonstrated anticancer effect against breast cancer and gastric cancer cells via modulation of ERK1/2 activity [202,203]. Although the efficacy of BCI in GBM treatment is also prospected, accumulation of further evidences would be necessary for therapeutic application.

### 3.5. PRL-3 Inhibitors

Inhibitors targeting PRL-3 have been developed as anti-cancer agents for years. Thienopyridone, a classical PRL-3 inhibitor targeting active site of PRL-3, blocks PRLs activity and has demonstrated anticancer effects [204]. In addition, Cmpd-43, a PRZ-3 inhibitor which interferes with PRL trimerization and possesses more specificity to PRZs, demonstrates anticancer activity against melanomas in vitro and in the xenograft model [205], and recently-developed monoclonal anti-PRL-3 antibody also exhibited anticancer activity in vitro and in vivo [206]; however, the effectiveness of these PRL-3 inhibitors in GBM treatment is not confirmed yet.

### 3.6. CDC25s Inhibitors

CDC25s play fundamental roles in regulation of GBM biology, and pharmacological inhibition of CDC25B or CDC25C expression resulted in growth arrest of GBM cells [170,172,173]. In addition, FDI-6, a recent developed small molecule inhibitor of FOXM1 which transcriptionally upregulates CDC25B expression, suppressed CDC25B expression in breast and ovarian cancer cells [207]. Although these CD25B or CDC25C inhibitors do not directly affect these phosphatases, the possibility of CDC25B or CDC25C as the pharmacological therapeutic target of GBM is suggested. Further development of direct specific inhibitors of these CDC25s would be necessary for GBM treatment.

### 3.7. EYA2 Inhibitor

As noted above, EYA2 is overexpressed and regulates growth and invasion of high-grade gliomas including GBMs [183]. MLS000544460, an allosteric Eya2 inhibitor, has been developed and inhibited migration of human epithelial cell line [215], and further investigation about effectiveness of MLS000544460 in GBM treatment is awaited.

Given current knowledge, it still remains difficult to determine which PPPs might be appropriate as a therapeutic target for GBM. Therefore, it is necessary to verify the effect and usefulness of PPP inactivation or activation in vitro before the efficacy of targeting PPPs which can be studies in in vivo models or clinical studies. As for inhibitors or activator of PPPs, substrate specificity and blood–brain barrier (BBB) permeability might pose issues, especially in the case of GBMs [216]. Now, as noted above, phase 2 clinical trial of PP2A inhibitor LB-100 against recurrent GBM cases has been already ongoing (https://clinicaltrials.gov/ct2/show/NCT03027388), and the drugs targeting PPPs with greater permeability of the BBB, such as SMAPs, have been developed recently [41,187,188]; clinical application of these agents in GBM therapy in the near future would be anticipated.

## 4. Conclusions

Recent studies on glioma biology have indicated the roles of PPPs in the regulation of GBM oncogenicity, and the modulation of PPP expression or activity for clinical GBM treatment is gradually getting closer. However, further in-depth studies are needed before the application of PPP modulation in clinical GBM treatment can be contemplated, as PPPs are more difficult as therapeutic targets than certain other molecules because a single PPP regulates multiple and complex signaling pathways. However, from another angle, PPP modulation might avoid relapse and the development of therapeutic insensitivity in GBM exactly because PPPs modulate diverse intracellular signaling pathways. In any case, targeting of PPPs in GBM therapy would be challenging but is worth further development. In future, highly specific and blood–brain barrier-permeable PPP inhibitors might have potential as novel therapeutic agents against refractory GBMs.

## Figures and Tables

**Figure 1 cancers-11-00241-f001:**
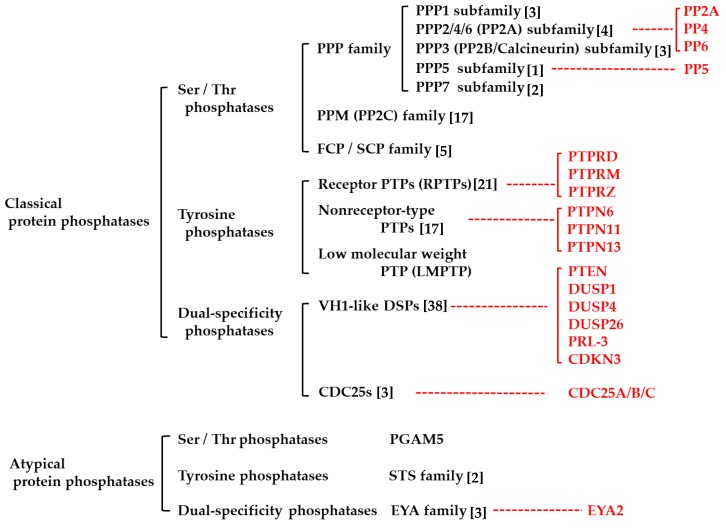
Protein phosphatases in human. Human phosphoprotein phosphatases (PPPs) are classified into two large classes as classical or atypical PPPs. And, these classes of PPPs are further grouped into families or subfamilies based on the amino-acid residue they dephosphorylate or their chemical structure. The PPPs focused in the text are written in red characters. Ser, Serine; Thr, Threonine; Tyr, Tyrosine; PPP, phosphoprotein phosphatase; PP2A, protein phosphatase 2A; PP2B protein phosphatase 2B; PP2C, protein phosphatase 2C; PPM, metal-dependent protein phosphatase; FCP, TFIIF-associating component of RNA polymerase Ⅱ carboxy-terminal domain phosphatase; PTP, protein Tyr phosphatase; LMPTP, low molecular weight PTP; VH1-like DSP [38], Vaccinia virus gene H1-like dual specificity phosphatase; PGAM5, phosphoglycerate mutase family member 5; Sts, suppressor of T-cell receptor signaling; EYA, eyes absent.

**Figure 2 cancers-11-00241-f002:**
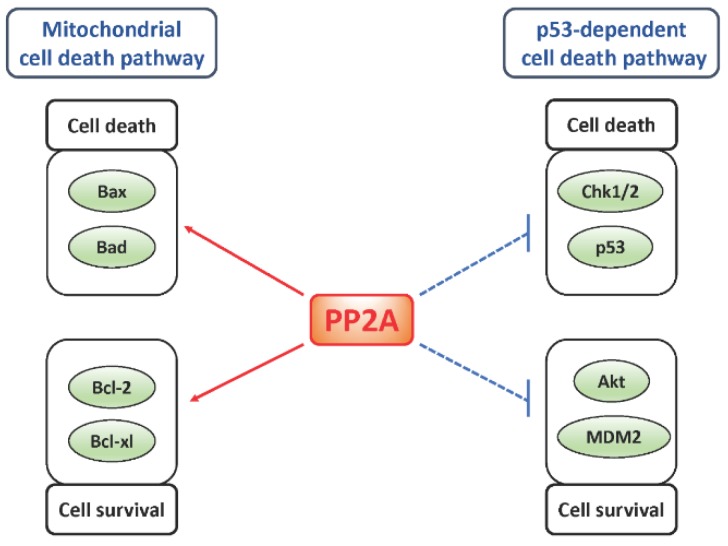
Multiple roles of protein phosphatase 2A (PP2A) in the regulation of cell death signaling. PP2A regulates both cell death-inducing and cell survival-inducing signaling simultaneously in not only mitochondria- but p53-dependent cell death cascade. In mitochondria-dependent cell death cascade, PP2A activates both pro-apopototic (Bax and Bad) and anti-apoptotic (Bcl-2 and Bcl-xl) Bcl-2 family proteins simultaneously by dephosphorylation. On the other hand, in p53-dependent cell death cascade, PP2A suppresses p53 inhibitor MDM2 and MDM2 activator Akt as well as p53 and p53 activator Chk1/2. Bcl-2, B cell lymphoma 2; Bcl-xl, B-cell lymphoma extra large; p53, tumor protein p53; Chk1/2, Serine/threonine-protein kinase Chk1/2; MDM2, Mouse double minute 2 homolog; Akt, protein kinase B.

**Table 1 cancers-11-00241-t001:** Summary of pharmacological inhibitors and activators of protein phosphatases highlighted in the text. The number of relevant references (Ref), molecular targets or mechanisms, and other information, such as clinical trial of each agent, are also shown. The agents with anti-glioblastoma (GBM) effects (and the number of relevant references) and clinical trial in GBM cases are emphasized in red characters.

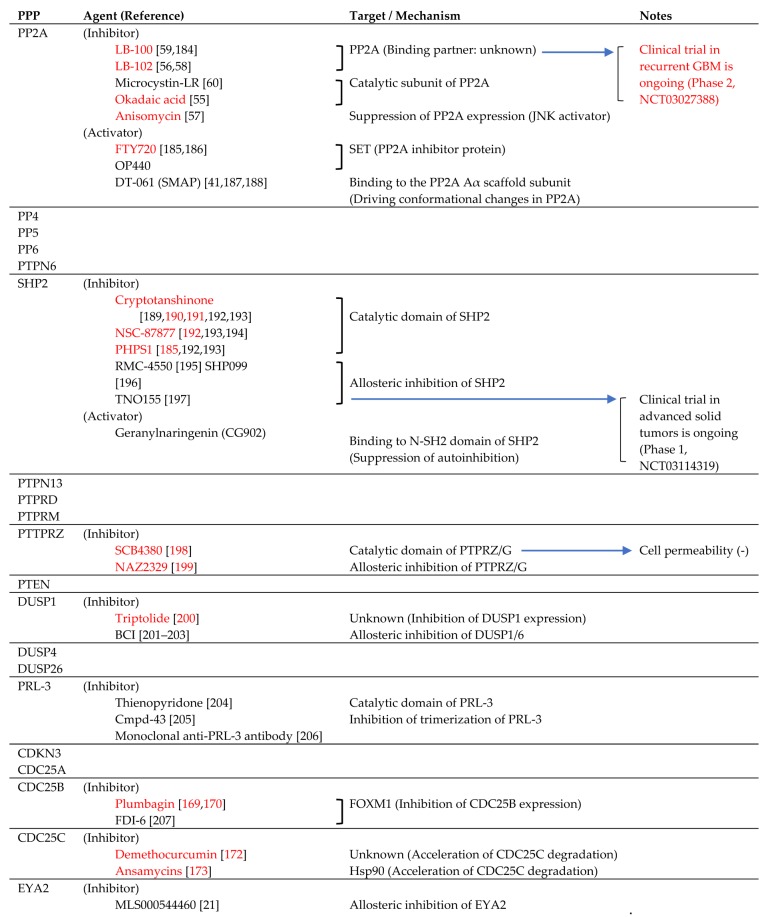

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
