# Peer review of "Protein Phosphatases—A Touchy Enemy in the Battle Against Glioblastomas: A Review"

_cancers, 2019, doi:10.3390/cancers11020241_

Round 1

Reviewer 1 Report

Glioblastoma is the highest grade and most common type of malignant brain tumor among adults. The average survival time is only 12-18 months. Thus, it is urgent to develop new drugs, treatment strategies and tools. In this review manuscript, the authors summarize those protein phosphatases which have been suggested to play important roles in the regulation of signal transductions in glioblastoma, and discuss their potential as therapeutic targets. Overall, it was written logically and appropriately, and would attract the attention from researchers who are interested in studying targeting protein phosphatases to treat glioblastoma.  

There are several issues which need to be addressed or corrected.

1. The subtitles can be better organized. Atypical protein phosphatases is one of the two classes of protein phosphatases, while its numbering is 2.4 and parallel to 2.1 PSPs, 2.2 PTPs and 2.3 DUSPs which are subgroups of classical protein phosphatases. It can be, for instance: 2. Protein phosphatases – 2.1 Classical protein phosphatases – 2.2 Atypical protein phosphatases.

2. Both Figure 1 and 2 look fuzzy. Better pictures are required. And the figure itself doesn’t have to be labeled as “Figure 1” or “Figure 2” at the lower-right corner.

3. In the Abstract, “the prognosis remains still poor, mainly because difficulty in gross total resection” is supposed to be “the prognosis still remains poor, mainly because of difficulty in gross total resection”.

4. In the Introduction, the origin of glioblastoma could be more specific.

5. In the Introduction, “Regulation of molecular signaling, including signaling regulated to tumor biology”. Is it supposed to be “related to”?

6. The font of subtitles should be identical. For instance, “2.1 Protein serine/threonine phosphatases”, “2.1.2. Protein phosphatase 4” and “2.1.3. Protein phosphatase 6” look strange.

7. In 2.1.1, “suggesting a role of PP2A as a tumor repressor”. “Tumor suppressor” is a more appropriate description.

8. In the Figure 2 legend, the font of “Mouse double minute 2 homolog” looks different.

9. In 2.1.2, “but they have different regulated”. Are there words missed?

10. “2.1.3. Protein phosphatase 6” should be “2.1.4. Protein phosphatase 6”.

11. In 3. PPPs as therapeutic targets, “As a current stage” is supposed to be “At the current stage”.

Author Response

Reviewer1

Glioblastoma is the highest grade and most common type of malignant brain tumor among adults. The average survival time is only 12-18 months. Thus, it is urgent to develop new drugs, treatment strategies and tools. In this review manuscript, the authors summarize those protein phosphatases which have been suggested to play important roles in the regulation of signal transductions in glioblastoma, and discuss their potential as therapeutic targets. Overall, it was written logically and appropriately, and would attract the attention from researchers who are interested in studying targeting protein phosphatases to treat glioblastoma.  

There are several issues which need to be addressed or corrected.

1.     The subtitles can be better organized. Atypical protein phosphatases is one of the two classes of protein phosphatases, while its numbering is 2.4 and parallel to 2.1 PSPs, 2.2 PTPs and 2.3 DUSPs which are subgroups of classical protein phosphatases. It can be, for instance: 2. Protein phosphatases – 2.1 Classical protein phosphatases – 2.2 Atypical protein phosphatases.

→ Thank you for important suggestion. We re-numbered and re-ordered the subtitles as #rReviewer1 suggested.

2.     Both Figure 1 and 2 look fuzzy. Better pictures are required. And the figure itself doesn’t have to be labeled as “Figure 1” or “Figure 2” at the lower-right corner.

→ Thank you for critical comments. As #Reviewer1 pointed out, we misformatted each figure. Therefore, we replaced all the figures for new figures with higher resolution and also removed the labels. 

3.     In the Abstract, “the prognosis remains still poor, mainly because difficulty in gross total resection” is supposed to be “the prognosis still remains poor, mainly because of difficulty in gross total resection”.

→ Thank you for the comment about details. We changed the sentence as #Reviewer1 suggested (page2 line4).

4.     In the Introduction, the origin of glioblastoma could be more specific.

→ Thank you for critical comments. We added more information about the origin of glioma (glioblastoma) (page3 line2-5).

5.     In the Introduction, “Regulation of molecular signaling, including signaling regulated to tumor biology”. Is it supposed to be “related to”?

→ Thank you for the critical comments. The suggestion of #Reviewer1 is right, and we changed the sentence as #Reviewer1 suggested (page3 line16).

6.     The font of subtitles should be identical. For instance, “2.1 Protein serine/threonine phosphatases”, “2.1.2. Protein phosphatase 4” and “2.1.3. Protein phosphatase 6” look strange.

→ Thank you for important comments. These fonts of subtitles were totally corrected.

7.     In 2.1.1, “suggesting a role of PP2A as a tumor repressor”. “Tumor suppressor” is a more appropriate description.

→ Thank you for important suggestion. The suggestion of #Reviewer1 is right, and we changed the sentence as #Reviewer1 suggested (page5 line10).

8.     In the Figure 2 legend, the font of “Mouse double minute 2 homolog” looks different.

→ Thank you for critical comments. These fonts of subtitles were totally corrected.

9.     In 2.1.2, “but they have different regulated”. Are there words missed?

→ Thank you for critical comments. As #Reviewer1 pointed out, the sentence missed the words. Therefore, we arranged the words and completed the sentence (page6 line10).

10.  “2.1.3. Protein phosphatase 6” should be “2.1.4. Protein phosphatase 6”.

→ Thank you for critical comments. Not only “Protein phosphatase 6” but all the subtitles were re-arranged and corrected.

      11. In 3. PPPs as therapeutic targets, “As a current stage” is supposed to be “At the current stage”.

     → Thank you for important suggestion. The suggestion of #Reviewer1 is right, and we changed the sentence as #Reviewer1 suggested (page18 line5).

Reviewer 2 Report

In their review submitted to Cancers, Tomiyama et al. provide an update of the current knowledge on protein phosphatases in glioblastomas. Unfortunately, the review is written rather superficially and nonsystematically. Compared to, for example, review by Navis et al. on tyrosine phosphatases in GBM (Navis at al. 2010 Acta Neuropathol), authors do not add much to the phosphatase field in GBM. In addition of Navis’ review, the authors only added Ser/Thr phosphatases, however, the recent studies highlighting following phosphatases in GBM biology are not covered, including PTPN12/PTP-PEST (Chen et al., 2018 Cancer Res), PTPN3 (Wang et al., 2018 Med Sci Monit), DUSP6 (Messina et al., 2011 Oncogene). Moreover, no comprehensive summary was provided considering latest genome data, or recently discovered small-molecule targeting phosphatases, with some of them being already in clinical trials for GBM and other cancer.

Main points:

1.     The review contains redundant and very generic conclusions that the phosphatases “would be suggested as meaningful” and “such information might have potential” for development of novel therapeutic agents. After reading a review the reader would want to have an additional part summarizing latest genome data and core pathways in GBM, with the roles the reviewed phosphatases playing in the pathways, as well as defining the most promising ones considering available phosphatase-targeting agents (see comments below). Here it is also suggested to make Figure 1 more GBM oriented, at least highlight the described PPPs.

 2.     In PPPs as therapeutic section, no evaluation of recently discovered small-molecule inhibitors and activators of phosphatases is provided. My suggestion is that authors have to modify this section and review available drugs targeting phosphatases paying special attention to clinical trials data and possible application for GBM therapy. At least SHP-2 inhibitors - SHP099, NSC-87877, cryptotanshinone; SHP-2 activator – geranylnaringenin; DUSP1 inhibitor – BCI; PTPRZ inhibitor - SCB4830; PP2A inhibitors - LB100 and LB102; PP2A activators - FTY720, and SMAPs (e.g. DT-061) should be reviewed. The phosphatase-targeting drugs were already comprehensively reviewed (Mazhar S et al., 2019 Biochim Biophys Acta Mol Cell Res; Frankson et al., 2017 Cancer Res., and Dedobbeleer et al. 2017 Biochemical Journal). Importantly, that LB100 is already in clinical trials (for example, see https://clinicaltrials.gov/ct2/show/NCT03027388) for GBM. In view of all this, the summary of the most promising phosphatases for targeting in GBM is required as a separate section or a table.

3.     In general, phosphatase inhibitors are mentioned in the text. However, authors avoid providing the names. This fact makes reading difficult and leads to misunderstanding, it is not clear if they talk about the same inhibitors or not. It has to be, for example, okadaic acid in Ref. 45 and LB100 in Ref 46 and so on.

4.     The PP2A section is written too superficially to cover the complexity of PP2A as a major Ser/Thr phosphatase. Authors should include more information about (i) PP2A structure (⁓60 heterotrimeric complexes with different substrate specificity are formed), (ii) PP2A regulation by endogenous PP2A inhibitory proteins, particularly PME-1, CIP2A and SET in GBM as well as (iii) strategies of PP2A targeting. Moreover, it was recently shown that PP2A regulates inhibitor resistance mechanism in cancers (Allen-Petersen et al., 2019 Cancer Res; Kauko et al., 2018 Sci Transl Med). Inhibition of the endogenous PP2A inhibitor, PME-1, also significantly sensitized glioblastoma cells to kinase inhibitors like MEK inhibitor, sunitinib, and PI3K inhibitor, LY294002 (Kaur at al., 2016 Cancer Res). Thus, there are plenty of studies showing that PP2A activity promote development of GBM and therapeutic resistance.

5.     Authors wrote “PTPRD is encoded by the PTPRD gene on chromosome 9p23–24.1 and is frequently inactivated in high-grade glioma cases accompanied with chromosome 9 deletion, which occurs frequently during the progression from low- to high-grade glioma [95].”

What does it mean “frequently”? Please indicate percentage, the same also for PP4C, PP6C, CDKN3, PTEN. Moreover, Ref.95 is referred to a quite old research, please use current genomic databases like cBioPortal.com to show relevant data. For example, PTPRD gene is deleted in 2-3% of the 532 GBM samples at cBioPortal.com instead of 40% of the 16 samples in Ref.95. 

Minor comments and suggestions:

1.     The current title does not provide a clear message of the review.

2.     Addition to Ref 123. PRL-3 was suggested as a valuable prognostic marker and is a promising therapeutic target for glioblastoma.

3.     It must be mentioned that LB100 leads to mitotic catastrophe in combination with DOX or TMZ treatment in GBM.

4.     LEOPARD syndrome with CAPITALS

5.     “PTPRD is encoded by the PTPRD gene on chromosome 9p23–24.1 and is frequently inactivated in high-grade glioma cases accompanied with chromosome 9 deletion, which occurs frequently during the progression from low- to high-grade glioma [95]”.

should be added “high-grade gliomas of the astrocytoma and oligodendroglioma types”.

6.     In general, the manuscript requires language editing and proofreading.

Author Response

#Reviewer2

In their review submitted to Cancers, Tomiyama et al. provide an update of the current knowledge on protein phosphatases in glioblastomas. Unfortunately, the review is written rather superficially and nonsystematically. Compared to, for example, review by Navis et al. on tyrosine phosphatases in GBM (Navis at al. 2010 Acta Neuropathol), authors do not add much to the phosphatase field in GBM. In addition of Navis’ review, the authors only added Ser/Thr phosphatases, however, the recent studies highlighting following phosphatases in GBM biology are not covered, including PTPN12/PTP-PEST (Chen et al., 2018 Cancer Res), PTPN3 (Wang et al., 2018 Med Sci Monit), DUSP6 (Messina et al., 2011 Oncogene). Moreover, no comprehensive summary was provided considering latest genome data, or recently discovered small-molecule targeting phosphatases, with some of them being already in clinical trials for GBM and other cancer.

Main points:

1.       The review contains redundant and very generic conclusions that the phosphatases “would be suggested as meaningful” and “such information might have potential” for development of novel therapeutic agents. After reading a review the reader would want to have an additional part summarizing latest genome data and core pathways in GBM, with the roles the reviewed phosphatases playing in the pathways, as well as defining the most promising ones considering available phosphatase-targeting agents (see comments below). Here it is also suggested to make Figure 1 more GBM oriented, at least highlight the described PPPs.

→ Thank you for critical comments. We added the information of the PPPs especially focused in the text by red character and also added the description in figure legend (page39 line7). 

 2.     In PPPs as therapeutic section, no evaluation of recently discovered small-molecule inhibitors and activators of phosphatases is provided. My suggestion is that authors have to modify this section and review available drugs targeting phosphatases paying special attention to clinical trials data and possible application for GBM therapy. At least SHP-2 inhibitors - SHP099, NSC-87877, cryptotanshinone; SHP-2 activator – geranylnaringenin; DUSP1 inhibitor – BCI; PTPRZ inhibitor - SCB4830; PP2A inhibitors - LB100 and LB102; PP2A activators - FTY720, and SMAPs (e.g. DT-061) should be reviewed. The phosphatase-targeting drugs were already comprehensively reviewed (Mazhar S et al., 2019 Biochim Biophys Acta Mol Cell Res; Frankson et al., 2017 Cancer Res., and Dedobbeleer et al. 2017 Biochemical Journal). Importantly, that LB100 is already in clinical trials (for example, see https://clinicaltrials.gov/ct2/show/NCT03027388) for GBM. In view of all this, the summary of the most promising phosphatases for targeting in GBM is required as a separate section or a table.

→ Thank you for critical comments. As #Reviewer2 suggested, we re-titled “PPPs as therapeutic target” section as “The phosphatase inhibitors and activators as the therapeutic agents of GBMs” and totally added and modified the contents of the section as #Reviewer2 suggested (page14 line31-page18 line8). In addition, the summary of the inhibitors and activators of PPPs focused in this session which might have therapeutic roles especially in GBMs was demonstrated as table1. (This revision is not enhanced and checked by the tools because the amount of revision is too much) 

3.     In general, phosphatase inhibitors are mentioned in the text. However, authors avoid providing the names. This fact makes reading difficult and leads to misunderstanding, it is not clear if they talk about the same inhibitors or not. It has to be, for example, okadaic acid in Ref. 45 and LB100 in Ref 46 and so on.

→ Thank you for important suggestion. As #Reviewer2 suggested, we added the name of each inhibitor which were not mentioned in the text (page5 line27, 29, 33, page10 line11, page 13 line2, 8).

4.     The PP2A section is written too superficially to cover the complexity of PP2A as a major Ser/Thr phosphatase. Authors should include more information about (i) PP2A structure (60 heterotrimeric complexes with different substrate specificity are formed), (ii) PP2A regulation by endogenous PP2A inhibitory proteins, particularly PME-1, CIP2A and SET in GBM as well as (iii) strategies of PP2A targeting. Moreover, it was recently shown that PP2A regulates inhibitor resistance mechanism in cancers (Allen-Petersen et al., 2019 Cancer Res; Kauko et al., 2018 Sci Transl Med). Inhibition of the endogenous PP2A inhibitor, PME-1, also significantly sensitized glioblastoma cells to kinase inhibitors like MEK inhibitor, sunitinib, and PI3K inhibitor, LY294002 (Kaur at al., 2016 Cancer Res). Thus, there are plenty of studies showing that PP2A activity promote development of GBM and therapeutic resistance.

→ Thank you for incisive comments. We totally arranged “PP2A” section (page4 line22-page6 line5) and added the information according to the comments of #Reviewer2. (This revision is not enhanced and checked by the tools because the amount of revision is too much)

5.     Authors wrote “PTPRD is encoded by the PTPRD gene on chromosome 9p23–24.1 and is frequently inactivated in high-grade glioma cases accompanied with chromosome 9 deletion, which occurs frequently during the progression from low- to high-grade glioma [95].”

What does it mean “frequently”? Please indicate percentage, the same also for PP4C, PP6C, CDKN3, PTEN. Moreover, Ref.95 is referred to a quite old research, please use current genomic databases like cBioPortal.com to show relevant data. For example, PTPRD gene is deleted in 2-3% of the 532 GBM samples at cBioPortal.com instead of 40% of the 16 samples in Ref.95. 

→ Thank you for critical suggestion. We added the information of the phosphatases about the frequency whether their expression or activity is genetically upregulated or downregulated in glioblastomas, from the cancer genome atlas (TCGA) (PP4C; page6 line14-15, PP6C; page7 line5-6, PTPRD; page8 line line35-36, PTEN; page10 line17-18, CDK3; page13 line7-8, CDC25s page13 line25-26, EYA2; page14 line23-24).

Reviewer 3 Report

In this review, a brief introduction and background of lists of protein phosphatases was included in the text. The information was helpful in promoting the understanding of the classification and function of common protein phosphatases. However, due to the complexity of their functions either as tumor suppressors or oncogenes, specific clinical implication is hard to make an easy conclusion. Therefore, the upstream regulators of protein phosphatase expression and/or activity might be of help. In current version of manuscript, the concepts of upstream regulators had been described, except that of microRNA. Therefore, it is recommended to have some information about the roles and effects of genetic and epigenetic regulation of protein phosphatases.

Author Response

 Reviewer3

In this review, a brief introduction and background of lists of protein phosphatases was included in the text. The information was helpful in promoting the understanding of the classification and function of common protein phosphatases. However, due to the complexity of their functions either as tumor suppressors or oncogenes, specific clinical implication is hard to make an easy conclusion. Therefore, the upstream regulators of protein phosphatase expression and/or activity might be of help. In current version of manuscript, the concepts of upstream regulators had been described, except that of microRNA. Therefore, it is recommended to have some information about the roles and effects of genetic and epigenetic regulation of protein phosphatases.

Thank you for critical suggestion. As #Reviewer3 suggested, the information about epigenetic or exosomal regulation pf PPPs in the text is not enough. Therefore, we added following two points.

1)    The knowledge about exosome-dependent regulation of PPPs in glioblastomas is very limited. So far as we concern, evidences about exosomal regulation of dual specificity phosphatase PTEN is only validated. Therefore, we added the information about exosome-mediated regulation of PTEN (page10 line33-page11 line4).

2)    AS the epigenetic regulation (=promoter methylation-dependent regulation) of PPPs in glioblastomas, protein tyrosine phosphatase SHP1 (PTPN6) is known to epigenetically downregulated in GBM cases. Therefore, we added about this information (page7 line16-line25).

Round 2

Reviewer 2 Report

Authors made a significant effort improving their manuscript, particularly shaping up its structure, and in therapeutic targeting part. However, I still have minor comments (the numbering follows the previous round comments)

(1)  I am very satisfied how the authors have modified the figure

(2)  Authors extensively covered phosphatase targeting section, however many of the molecules described in the text are missing from the table. This fact makes false impression on readers that only a limited progress was made in the field. It is highly recommended to complete the table by (i) listing the molecules mentioned in the text; (ii) noting whether compounds were tested against GBM (providing references); (iii) clinical trial numbers have to be indicated also in the table that will help readers to navigate.

I would not fully agree with author’s conclusion on page 11 lines 43-44 “At the current stage, blood brain barrier permeability of phosphatase inhibitors is still a major issue [22, 217], therefore, further improvement of these drugs might be necessary for clinical use”. There are well established compounds, including LB-100, perphenazine (antipsychotic drugs available in clinic for more than 40 years), and SMAPs (that are derivatives of antipsychotics), that cross BBB. Therefore, taking into account available inhibitors, clinical trials and BBB crossing properties, the most promising phosphatases for targeting GBM have to be noted.

 I strongly recommend to rewrite a title of the section, e.g. Phosphatase targeting in GBM.

Page 9 line 38, change “each” to “some”

Page 11 line 35, is numbered title of the section needed?

(3)  The authors fully addressed my previous comment.

(4)  The authors greatly expanded PP2A section. I do not have any more comments on that.

(5)  The authors fully addressed my previous comment.

Minor comments

 Choose only one way of writing “in vitro, in vivo, in vitro, in vivo, in-vitro”

Author Response

The details are described in the point-by-point response below. And, the changed or modified points of the text are enhanced by blue highlighter tool.

Responses to the #Reviewer2

(The changed or modified points of the text are enhanced by blue highlighter tool in the text.)

Reviewer2

(1)  Authors extensively covered phosphatase targeting section, however many of the molecules described in the text are missing from the table. This fact makes false impression on readers that only a limited progress was made in the field. It is highly recommended to complete the table by (i) listing the molecules mentioned in the text; (ii) noting whether compounds were tested against GBM (providing references); (iii) clinical trial numbers have to be indicated also in the table that will help readers to navigate.

→ Thank you for important suggestion. We modified table1 as suggested and also changed the legend of table1 along with modification (page33 line27-31).

I would not fully agree with author’s conclusion on page 11 lines 43-44 “At the current stage, blood brain barrier permeability of phosphatase inhibitors is still a major issue [22, 217], therefore, further improvement of these drugs might be necessary for clinical use”. There are well established compounds, including LB-100, perphenazine (antipsychotic drugs available in clinic for more than 40 years), and SMAPs (that are derivatives of antipsychotics), that cross BBB. Therefore, taking into account available inhibitors, clinical trials and BBB crossing properties, the most promising phosphatases for targeting GBM have to be noted.

→ Thank you for very important suggestion. We removed the sentence of page 11 line43-44 (“At the current stage, blood brain barrier permeability of phosphatase inhibitors is still a major issue [22, 217], therefore, further improvement of these drugs might be necessary for clinical use”) and re-wrote new conclusion (page11 line42 – page12 line1).

 I strongly recommend to rewrite a title of the section, e.g. Phosphatase targeting in GBM.

→ Thank you for sharp opinion. We changed the title as suggested (page9 line36).

Page 9 line 38, change “each” to “some”

→ Thank you for important indication. We changed the word as suggested.

Page 11 line 35, is numbered title of the section needed?

→ Thank you for important indication. We removed the section title as suggested.

(3)  The authors greatly expanded PP2A section. I do not have any more comments on that.

(5)  The authors fully addressed my previous comment.

Minor comments

 Choose only one way of writing “in vitro, in vivo, in vitro, in vivo, in-vitro

       → Thank you for pointing out the mistake. We unified writing as “in vitro”.